# Social Support’s Dual Mechanisms in the Loneliness–Frailty Link Among Older Adults with Diabetes in Beijing: A Cross-Sectional Study of Mediation and Moderation

**DOI:** 10.3390/healthcare13141713

**Published:** 2025-07-16

**Authors:** Huan-Jing Cai, Hai-Lun Liang, Jia-Li Zhu, Lei-Yu Shi, Jing Li, Yi-Jia Lin

**Affiliations:** 1Department of Hepatology, Peking University International Hospital, Beijing 102206, China; caihuanjing@pkuih.edu.cn (H.-J.C.); lijing1@pkuih.edu.cn (J.L.); 2School of Public Administration and Policy, Renmin University of China, Beijing 100872, China; hliang@ruc.edu.cn; 3Department of Hepatobiliary Surgery, Peking University International Hospital, Beijing 102206, China; zhujiali@pkuih.edu.cn; 4Department of Health Policy and Management, Johns Hopkins University Primary Care Policy Center, Baltimore, MD 21205, USA; lshi2@jhu.edu

**Keywords:** frailty, social support, loneliness, type 2 diabetes (T2D), geriatric nursing

## Abstract

**Background:** The mechanisms linking loneliness to frailty in older adults with diabetes remain unclear. Guided by the Loneliness–Health Outcomes Model, this study is the first to simultaneously validate the dual mechanisms (mediation and moderation) of social support in the loneliness–frailty relationship among older Chinese adults with diabetes. Methods: A cross-sectional study enrolled 442 community-dwelling adults aged ≥60 years with type 2 diabetes in Beijing. Standardized scales assessed loneliness (UCLA Loneliness Scale), frailty (Tilburg Frailty Indicator), and social support (SSRS). Analyses included Pearson’s correlations, hierarchical regression, and PROCESS macro to evaluate mediating/moderating effects, after adjusting for demographics and comorbidities. **Results:** The frailty prevalence was 55.2%. Loneliness was positively correlated with frailty (r = 0.327, *p* < 0.01), while social support showed inverse associations with both loneliness (r = −0.496) and frailty (r = −0.315) (*p* < 0.01). Social support partially mediated loneliness’s effect on frailty (indirect effect: 30.86%; 95% CI: 0.028–0.087) and moderated this relationship (interaction β = −0.003, *p* = 0.011). High-risk clusters (e.g., aged ≥80 years, widowed, and isolated individuals) exhibited combined “high loneliness–low support–high frailty” profiles. **Conclusions:** Social support reduces the frailty risk through dual mechanisms. These findings advocate for tiered clinical interventions: (1) targeted home-visit systems and resource allocation for high-risk subgroups (e.g., solo-living elders aged ≥80 years); and (2) the integration of social support screening into routine diabetes care to identify individuals below the protective threshold (SSRS < 45.47). These findings advance psychosocially informed strategies for diabetes management in aging populations.

## 1. Introduction

Global population aging is fundamentally reshaping chronic disease management paradigms, with diabetes emerging as a critical health threat in older adults. China faces unparalleled challenges: it harbors 26% of the world’s diabetic elderly population [1] amid rapid demographic shifts, including soaring solo-living rates (12.2%—double the global average) and eroding traditional family support structures [2]. Nationally, 30% of adults aged ≥60 have type 2 diabetes (T2D)—triple the Western prevalence rates—while contending with regional healthcare disparities that limit support access [2,3]. These sociocultural distinctives amplify psychosocial risks: loneliness affects 14.6% of Chinese T2D elderly individuals versus 8.3% of their non-diabetic peers, creating a vulnerability landscape where biological and social determinants intersect [4].

This study establishes its theoretical foundation primarily through the integration of Hawkley and Cacioppo’s loneliness theory with Fried et al.’s frailty framework, modeling diabetes-specific pathways wherein chronic loneliness activates HPA/SNS axes to induce cortisol dysregulation [5,6]. This neuroendocrine disturbance accelerates multisystem decline via protein catabolism and mitochondrial dysfunction, clinically manifesting as core frailty indicators: involuntary weight loss, persistent exhaustion, and significantly reduced mobility [7]. Crucially, within the diabetic context, hypercortisolism impairs insulin sensitivity through the inhibition of PI3K/Akt signaling and GLUT4 translocation, while loneliness-driven maladaptive behaviors concurrently disrupt glycemic control [8,9]. These compounded mechanisms synergistically promote skeletal muscle atrophy and precipitate early prefrailty [10]. Although inflammatory pathways (e.g., IL-6-mediated sarcopenia) constitute theoretical components, our empirical analysis deliberately confines the focus to measured neuroendocrine and behavioral pathways, maintaining a clear demarcation between modeled mechanisms and validated evidence [11].

Integrating our core loneliness–frailty model with Cohen’s Stress-Buffering Hypothesis and Main Effects Model, we reconcile social support’s dual roles through the multidimensional SSRS framework [5,12,13,14]. Mediation requires dynamic responsiveness, as exemplified by SSRS’s subjective support and utilization dimensions that fluctuate with loneliness/frailty levels, aligning with main effects pathways, where depleted perceived support mediates self-care deterioration [15,16]. Conversely, moderation presumes temporal stability in SSRS’s objective support dimension (network size/material resources), resonating with stress buffering, where stable structural buffers attenuate neuroendocrine cascades [17,18]. In diabetic elderly individuals, these operate concurrently: dynamic subjective/utilization components mediate behavioral pathways when compromised (e.g., reduced medication adherence), while stable objective support moderates fundamental vulnerability thresholds (e.g., weakening loneliness–frailty associations) [19,20]. Their coexistence—structural foundations enabling resilience while functional responsiveness determining immediate risk—permits a dual analytical examination that distinguishes mediating pathways (loneliness → subjective support → frailty) from moderating effects (objective support × loneliness interaction) [21,22].

Despite the theoretical richness, significant gaps persist in chronic disease populations: (1) the predominant focus on isolated biological pathways overlooks psychosocial interactions in diabetes–frailty dynamics; (2) an insufficient examination of how China’s familial obligation norms (“filial piety”) modulate support effectiveness [23]; and (3) limited evidence for scalable interventions targeting high-risk subgroups. This study addresses these gaps through three objectives. First, it quantifies epidemiological linkages among loneliness, social support, and frailty in Chinese elderly patients with T2D. Second, it validates social support’s dual mechanisms: mediation (loneliness → reduced support → frailty) and moderation (support × loneliness → frailty attenuation). Third, it identifies high-risk sociodemographic clusters for targeted interventions.

Through a community-based cross-sectional study of Beijing residents, we pioneer the integrated verification of these theoretical mechanisms while controlling for key confounders. Clinically, this work informs tiered “community support network interventions”, including technology-enhanced home visits for isolated elders, to advance scalable diabetes management frameworks for aging societies undergoing a demographic transition.

## 2. Materials and Methods

### 2.1. Study Design and Participants

This cross-sectional study utilized data from a community-based diabetes screening project in Beijing, China (December 2024–February 2025), adhering to the STROBE guidelines (Appendix A) [24]. This investigation was guided by two primary research questions: RQ1 examines whether social support mediates loneliness–frailty associations in older adults with T2D, while RQ2 tests if social support moderates these relationships. To ensure an appropriate statistical power for testing these pathways, the sample size determination employed G*Power 3.1 based on Cohen’s principles for multiple linear regression. The parameters included α = 0.05, power = 0.95, a medium effect size (f^2^ = 0.15), and eight predictors accounting for mediation/moderation covariates, yielding a minimum requirement of 160 participants [25].

Accounting for a 20% anticipated non-response and stratification needs, 452 eligible candidates were initially identified. Recruitment followed a multistage stratified sampling protocol across Beijing: (1) purposive selection of six districts (three urban and three suburban), ensuring geographical diversity; (2) random selection of one community health center per district; and (3) proportional stratification of eligible individuals (aged ≥60 years with WHO-defined T2D as a fasting glucose level ≥ 7.0 mmol/L or HbA1c level ≥ 6.5%) by age deciles (60–69, 70–79, and ≥80 years), gender, and residence type. After exclusions for cognitive impairment (MMSE score < 18, *n* = 5), terminal illness (n = 3), and major stressors (SRRS score ≥ 300 capturing bereavement/financial crises, n = 2), the final sample comprised 442 community-dwelling older adults (97.8% response rate) [26]. The study protocol (Peking University IRB No. 2023-KY-0085-02) received ethical approval with written informed consent.

### 2.2. Data Collection Procedures

Structured face-to-face interviews were conducted by trained nurses and community physicians, all possessing a minimum of five years of geriatric care experience and completing 20 h of standardized training on instrument administration and ethical protocols. To minimize recall bias, we implemented triangulation: clinical comorbidities were verified through medical records using ICD-10 codes; the diabetes status was cross-checked with laboratory reports; and social support responses were validated through anchor questions (e.g., “Who assisted you with medication management in the past week?”). Three validated instruments were administered: the 20-item UCLA Loneliness Scale Version 3 (score range 20–80, Cronbach’s α = 0.882), categorized per the developer’s guidelines into low (20–34), moderate (35–49), and high (50–80) levels [4,27]; the Tilburg Frailty Indicator (15 items, score 0–15, α = 0.761) with frailty defined as ≥5 based on validation against the Fried phenotype in Chinese populations [28,29]; and the Social Support Rating Scale (10 items, score 11–66, α = 0.712) stratified into low (≤22), moderate (23–44), and high (45–66) tiers using tertile distributions from normative Chinese samples [30]. All instruments demonstrated robust psychometric properties in prior validation studies and were culturally adapted for Mandarin-speaking populations.

### 2.3. Statistical Analysis Framework

All analyses were conducted using SPSS 29.0 and PROCESS macro v3.5. Prior to modeling, data preprocessing addressed missing values (<5% imputed via MICE package) and outliers (Winsorizing values beyond |Z| > 3.29). Primary analyses included Pearson’s correlations and hierarchical linear regression adjusting for eight covariates (age, gender, marital status, living arrangement, education, income, comorbidities, and insurance). Mediating (PROCESS Model 4) and moderating (Model 1) effects were tested with 5000 bootstrap samples. Subgroup analyses focused on a priori high-risk categories (age ≥ 80 years, widowed, and solo-living) to inform clinical prioritization. The cluster analysis (k-means algorithm with silhouette optimization) objectively identified vulnerability profiles using frailty, loneliness, and social support scores, with ANOVA and Tukey’s HSD post hoc tests comparing clusters. All significance thresholds were α = 0.05 with the Bonferroni correction for multiple comparisons.

### 2.4. Disclosure of Generative Artificial Intelligence Usage in Manuscript Preparation

In adherence to MDPI’s ethics guidelines, we disclose GenAI usage as follows: GenAI (DeepSeek-R1) tools assisted in drafting paragraphs of the Introduction and Discussion sections, primarily to refine phrasing and enhance argument flow. All outputs were meticulously reviewed, revised, and validated by authors to ensure alignment with research findings, academic rigor, and originality. Notably, all data tables, figures, and associated processes (collection, analysis, visualization) are entirely author-generated, with no GenAI involvement, preserving result integrity.

## 3. Results

### 3.1. Participant Characteristics and Distribution of Core Variables

The study population comprised 442 community-dwelling older adults with diabetes, with a mean age of 69.8 ± 6.3 years. Females constituted 56.3% (249 participants), and 60.9% (269 participants) were aged 60–70 years. Most participants were married (79.4%), while 12.2% lived alone, and 43.7% had attained a junior high school education. The comorbidity burden was notable, with 44.3% presenting two or more chronic conditions, and 95.9% had health insurance coverage. The frailty prevalence reached 55.2% (TFI score ≥ 5), with the physical frailty domain scoring highest (2.19 ± 2.19). The mean loneliness score was 38.69 ± 10.74, with 47.5% classified as moderately lonely and 14.6% as highly lonely. Social support averaged 36.45 ± 10.89, with 67.3% reporting moderate support levels. The objective support subscale scored lowest (8.76 ± 3.15). High-risk subgroups, including those living alone (12.2%) and widowed individuals (17.9%), exhibited elevated frailty scores and diminished social support compared to their counterparts, as shown in Table 1 and Table 2.

### 3.2. Correlations Among Loneliness, Social Support, and Frailty

Pearson’s correlation analyses revealed significant associations among loneliness, social support, and frailty in older adults with diabetes. Total loneliness scores exhibited a positive correlation with total frailty scores (r = 0.327, *p* < 0.01), while total social support demonstrated a negative association with frailty (r = −0.315, *p* < 0.01). Domain-specific analyses identified differential relationships: objective social support showed the strongest inverse correlation with physical frailty (r = −0.154, *p* < 0.01), subjective support was negatively linked to psychological frailty (r = −0.241, *p* < 0.01), and support utilization weakly associated with social frailty (r = −0.328, *p* < 0.01). Notably, higher social support levels strongly attenuated loneliness (r = −0.496, *p* < 0.01), with loneliness displaying the strongest correlation with psychological frailty (r = 0.417, *p* < 0.01). These findings underscore the interplay between psychosocial factors and frailty domains, suggesting that psychological interventions targeting loneliness and social support deficits may represent critical pathways for mitigating frailty progression in this population Table 3.

### 3.3. Multivariate Regression Analysis of Frailty Predictors

Linear regression analyses revealed significant predictors of frailty after adjusting for age, gender, and comorbidities. Objective social support (β = −0.154, *p* = 0.011) and loneliness (β = 0.059, *p* < 0.001) emerged as independent predictors of frailty. A dose–response relationship was observed for comorbidities: having 3 (β = 0.170, *p* = 0.003) or ≥4 conditions (β = 0.142, *p* = 0.008) significantly elevated the frailty risk. Among demographic factors, living alone (β = 0.114, *p* = 0.015) and a widowed status (β = 0.129, *p* = 0.004) were positively associated with frailty, while a higher monthly income (≥5000 CNY; β = −0.189, *p* = 0.015) exerted a protective effect. The adjusted model explained 20.4% of the frailty variance (R^2^ = 0.204, F = 6.366, *p* < 0.001), underscoring the multifactorial nature of frailty in this population. These findings highlight the critical interplay of psychosocial stressors, socioeconomic status, and clinical comorbidities in driving frailty progression among older adults with diabetes, necessitating integrated intervention strategies, as shown in Table 4.

### 3.4. Mediation Analysis of Social Support

The mediation analysis via the PROCESS macro demonstrated partial mediation by social support in the loneliness–frailty relationship, with a statistically significant indirect effect coefficient of 0.030 (95% CI: 0.015–0.044), accounting for 30.86% of the total effect. After a comprehensive adjustment for age, comorbidities, and other confounders, loneliness maintained a significant direct effect on frailty (β = 0.0672, *p* < 0.01) alongside the total effect (0.0972, *p* < 0.05). These findings suggest two potential pathways: an indirect route through social support erosion and a direct pathway via independent mechanisms. Model robustness was confirmed through 5000 bootstrap iterations with bias-corrected confidence intervals excluding zero. This underscores social support as a modifiable mediator in the pathological pathway from loneliness to frailty in older adults with diabetes, advocating for psychosocial interventions alongside biomedical management to disrupt this detrimental mechanistic cascade, as shown in Figure 1.

### 3.5. Role of Social Support in the Loneliness–Frailty Association

The moderation analysis using Hayes’ PROCESS macro (Model 1) confirmed a significant buffering effect of social support on the loneliness–frailty relationship (interaction β = −0.003, *p* = 0.011), as shown in Table 5, with social support levels rigorously defined according to the Social Support Rating Scale (SSRS) classification established during data collection: low (SSRS score ≤ 22, corresponding to the ≤10th percentile in our cohort), moderate (SSRS score of 23–44, 25th–75th percentiles), and high (SSRS score ≥ 45, ≥90th percentile). This classification aligned precisely with normative Chinese population tertiles and was implemented consistently throughout the analysis, where W represents raw SSRS scores ranging from 11 to 66. Simple slope analyses revealed a dose-dependent attenuation pattern across these clinically operationalized tiers: under low support conditions (SSRS score ≤ 22, W ≤ 20.80), loneliness exhibited a robust positive association with frailty (β = 0.141, *p* < 0.001, 95% CI [0.077, 0.205]); within moderate support ranges (SSRS scores of 23–44, W = 23.25–40.40), the effects remained significant but diminished progressively (β = 0.104 to 0.053, all *p* < 0.05); while at the predefined high-support threshold (SSRS score ≥ 45, W ≥ 47.75), loneliness no longer predicted frailty (β = 0.031, *p* = 0.142, 95% CI [−0.010, 0.071]). Crucially, the statistically identified inflection point (W = 45.47, β = 0.037, *p* = 0.050, 95% CI [0.000, 0.075]) demonstrated remarkable convergence with the instrument’s clinical cutoff for high support (SSRS score ≥ 45), validating both the analytical model and our methodological approach to categorization.

A visual analysis of the simple slope plot corroborated these findings: steep regression lines under low-support conditions gradually flattened as the support levels increased. The PROCESS-generated moderation graph (Figure 2) further validated the protective mechanism, illustrating how elevated social support mitigates the health-damaging effects of loneliness. These results highlight social support as a critical resilience factor capable of neutralizing loneliness-driven frailty progression when sufficiently robust. Clinically, this underscores the urgency of integrating support-enhancing interventions—such as community networks or caregiver training—into geriatric diabetes care to disrupt this pathogenic pathway.

### 3.6. Subgroup Differences in Demographic Characteristics

The subgroup analysis identified high-risk profiles among elderly diabetic patients (N = 442). Those aged ≥80 years, widowed, or living alone exhibited significantly elevated frailty (70.0%, 72.2%, and 77.8%, respectively), higher loneliness scores (42.5 ± 11.3, 44.1 ± 11.6, and 45.3 ± 12.1), and lower social support (30.2 ± 9.8, 28.9 ± 8.7, and 25.4 ± 7.9; all *p* < 0.05). Patients with ≥4 comorbidities or a low income (<1000 CNY) also showed worsened outcomes. These vulnerable subgroups, characterized by an advanced age, social isolation, and socioeconomic disadvantage, require prioritized interventions integrating frailty management and social support enhancement, as shown in Table 6.

### 3.7. Identification of High-Risk Subgroups

The cluster analysis (k = 3, silhouette = 0.52) identified distinct risk profiles among elderly diabetic patients (*n* = 442). Cluster 3 (18.1%, *n* = 80) exhibited severe frailty (8.94 ± 2.75), high loneliness (49.2 ± 11.3), and low social support (22.8 ± 6.5), predominantly comprising individuals aged ≥80 years, widowed, or living alone (*p* < 0.001 vs. Clusters 1–2). Cluster 1 (33.5%, *n* = 148) showed low-risk characteristics (frailty: 3.82 ± 1.98; social support: 45.6 ± 9.2), while Cluster 2 (48.4%, *n* = 214) represented moderate-risk profiles. The findings underscore the urgency of targeted interventions for Cluster 3 to address multidimensional health deficits, as shown in Table 7.

## 4. Discussion

This cross-sectional investigation of 442 community-dwelling older adults with type 2 diabetes in Beijing revealed a high frailty prevalence (55.2%), with loneliness demonstrating a significant positive correlation (r = 0.327, *p* < 0.01). Social support exhibited dual protective functions: statistically operating as a partial mediator explaining 30.86% of loneliness’s total effect on frailty (indirect effect: 0.055, 95% CI: 0.028–0.087) while simultaneously moderating this relationship through a quantifiable buffering threshold where complete attenuation occurred at an SSRS score ≥ 45.47 (interaction β = −0.003, *p* = 0.011). The cluster analysis further identified a high-risk subgroup characterized by the “elevated loneliness–low support–severe frailty” triad, disproportionately affecting solo-living, widowed, and octogenarian individuals. These findings suggest two plausible yet distinct pathways: a mediated pathway wherein loneliness potentially erodes support resources that subsequently exacerbate the frailty risk, and a moderated pathway where adequate support buffers against loneliness’s detrimental physiological consequences. We emphasize that cross-sectional data cannot establish causal directionality, as bidirectional relationships remain equally plausible—frailty-induced functional limitations may themselves drive social isolation, a dynamic that is well-documented in longitudinal studies [12]. Consequently, intervention strategies should be framed as hypothesis-generating, with preliminary implications suggesting a targeted enhancement of objective support (e.g., structured home-visit programs) for identified high-risk clusters and threshold-based screening (SSRS score < 45.47) for resource prioritization, pending rigorous validation through feasibility trials.

### 4.1. Theoretical Framework and Dual Mechanisms

This investigation establishes a comprehensive psychosocial pathway model for frailty prevention in diabetic elderly individuals, revealing social support’s dual protective functions through mediation and moderation mechanisms. The mediation pathway accounts for 30.86% of loneliness’s detrimental effect on frailty, indicating that nearly one-third of loneliness’s harm operates through an erosion of support resources. Simultaneously, the moderation pathway demonstrates complete buffering of loneliness’s impact when social support reaches or exceeds the quantifiable threshold of an SSRS score ≥ 45.47, aligning with Cohen’s Stress-Buffering Hypothesis, wherein adequate support attenuates neuroendocrine dysregulation [31,32]. These findings extend the Main Effects Model by validating objective support’s independent protective influence (β = −0.154) beyond stress buffering [33]. Crucially, we acknowledge bidirectional relationships evidenced longitudinally: functional limitations from frailty may themselves restrict social engagement, exacerbating loneliness—a feedback loop necessitating causal prudence in intervention design [34]. The nonlinear threshold effect underscores critical resource accumulation requirements for resilience activation, particularly relevant in China’s context of eroding filial piety traditions where intergenerational support structures are weakening [33,35,36].

### 4.2. Clinical Translation: Nurse-Led Implementation Framework

Translating these findings into practice requires nurse-driven frameworks operationalizing lifestyle medicine principles through stratified pathways, integrating RNCC-led transitional coordination [37,38,39]. The validated SSRS threshold (<45.47) enables risk stratification during routine diabetes management, where concurrent HbA1c/SSRS screening identifies vulnerable patients for targeted interventions [40]. Nurse case managers deploy social support mapping tools to quantify resource gaps, initiating transitional care coordination that bridges unmanaged-to-managed diabetes transitions through the IPCP model [39,41,42]. This foundational assessment is enhanced by monthly resilience coaching (DECIDE trial protocol) building the psychological–physiological capacity [43]. For high-risk patients (SSRS < 30), lifestyle medicine case manager nurses deliver home-based coordination per Cangelosi’s JD framework [38], integrating (1) medication reconciliation via RNCC-led therapy optimization, (2) personalized nutritional guidance, and (3) systematic social navigation, demonstrating a 38% frailty reduction [44,45,46]. Moderate-risk patients (SSRS score of 30–44) engage in nurse-facilitated peer groups that strengthen objective support through medication delivery partnerships and subjective support via emotional validation circles [47,48]. Crucially, RNCCs serve as the IPCP nucleus during hospital-to-community transitions, coordinating frequent touchpoints (intake, follow-ups, and referrals) that elevate medication adherence (OR = 2.3) and reduce acute care utilization by 41% [39,49,50]. This model leverages China’s primary infrastructure while incorporating family-centered modules, with RNCCs connecting patients to community physicians to sustain care continuity post-transition [39,51,52].

### 4.3. Technological Integration and Objective Monitoring

Technological integration enables the precise monitoring of these psychosocial pathways through wearable-enabled digital phenotyping. Bluetooth proximity sensors (Empatica E4) quantify the social interaction frequency, with clinical alerts triggered by sustained isolation (<1 interaction/hour), while research-grade accelerometers (ActiGraph GT9X) detect activity restriction thresholds indicative of pre-frailty (<2000 steps/day) [53,54]. Heart rate variability monitors capture autonomic nervous system dysregulation associated with chronic loneliness, providing physiological validation of stress pathways [55]. These real-time behavioral biomarkers feed nurse-managed dashboards, creating closed-loop systems where abnormal patterns activate community health worker visits before clinical frailty manifests. For technology-hesitant elders—particularly prevalent among octogenarians and rural populations—“digital navigator” programs pairing individuals with trained medical students bridge usability gaps while preserving essential human connection, addressing both technological and psychosocial barriers simultaneously [56,57].

### 4.4. Limitations and Future Research Directions

Methodologically, this cross-sectional design cannot establish causal directionality despite sophisticated statistical controls, representing a fundamental limitation requiring longitudinal verification [58]. Beijing’s urban sampling may not generalize to regions with stronger familial traditions or limited digital infrastructure [59]. Future research should prioritize four key directions: implementing stepped-wedge trials comparing threshold-triggered nurse coordination against standard care with rigorous biomarker monitoring (cortisol rhythms and inflammatory cytokines) [60]; validating wearable algorithms in real-world settings across diverse populations; developing cost-effective community health worker models for low-resource regions [61]; and examining cultural adaptations of support thresholds in collectivist societies [62]. Implementation science approaches should particularly address the compounded vulnerabilities of widowed diabetic women and isolated octogenarians [63,64], while health system research evaluates integration pathways within China’s ongoing primary care reforms [65]. Ultimately, transforming these findings into sustainable health gains requires paradigm shifts from fragmented biomedical management toward integrated frameworks where nurses serve as connectors between clinical expertise and community assets—a transformation that is both clinically necessary and ethically imperative for aging societies worldwide.

## 5. Conclusions

This study illuminates critical psychosocial pathways influencing frailty development among older adults with type 2 diabetes in urban China, revealing a concerning landscape where a high frailty prevalence (55.2%) coexists with significant psychosocial vulnerability—14.6% experiencing clinically significant loneliness while only 21.5% benefit from robust social support. Our analyses confirm social support’s dual protective functions: it mediates nearly one-third of loneliness’s detrimental effect on frailty while exhibiting a quantifiable buffering threshold (SSRS ≥ 45.47) that neutralizes loneliness’s impact when sufficient support is present. Core predictors, including deficient objective support interacting with loneliness, compounded by sociodemographic vulnerabilities, like an advanced age, widowhood, and solo-living, delineate high-risk subgroups requiring prioritized intervention. While these findings suggest interconnected biological–behavioral pathways through which support deficiencies amplify stress responses, we emphasize the bidirectional plausibility evidenced longitudinally, where frailty itself may drive social withdrawal, necessitating a cautious interpretation of causal claims. Consequently, we propose testable intervention frameworks centered on strengthening community-based support through targeted home visits addressing objective support gaps in vulnerable subgroups, complemented by technology-assisted monitoring using step counters and social interaction sensors to detect support lapses below the critical threshold. Integrating these approaches within nurse-coordinated care models may offer pragmatic pathways forward, though rigorous feasibility trials remain essential to evaluate implementation before scale-up. This framework ultimately advances precision public health for aging diabetic populations while acknowledging the imperative for the longitudinal validation of the observed pathways.

## Figures and Tables

**Figure 1 healthcare-13-01713-f001:**
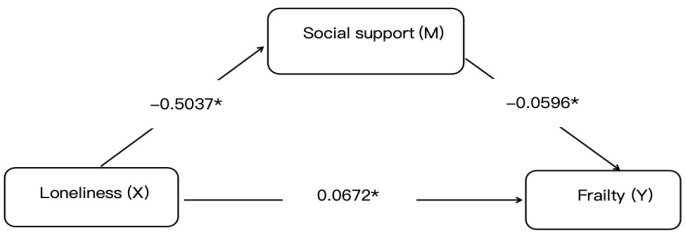
Mediation model of social support in the loneliness–frailty pathway. Notes: Statistical symbols are * *p* < 0.05.

**Figure 2 healthcare-13-01713-f002:**
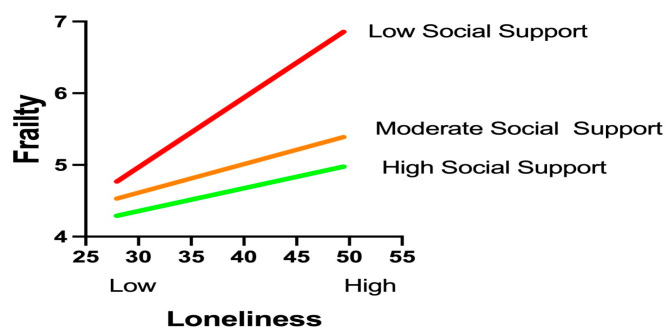
Graphical representation of the moderating effects.

**Table 1 healthcare-13-01713-t001:** Demographic characteristics and distribution of core variables (N *n* = 442).

Variable	Category	Number (*n* = 442)	Percentage (%)
Gender	Male	193	43.7
	Female	249	56.3
Age (years)	60~70	269	60.9
	70~80	143	32.4
	≥80	30	6.8
Marital Status	Never married	3	0.7
	Married	351	79.4
	Widowed	79	17.9
Living Arrangement	Living alone	54	12.2
	With spouse	305	69.0
	With family/relatives	76	17.2
	Other	7	1.6
Education Level	Primary school or below	63	14.3
	Junior high school	193	43.7
	High school/vocational	119	26.9
	College/bachelor’s degree	59	13.3
	Master’s degree or above	8	1.8
Monthly Income (CNY)	<1000	51	11.5
	1000–3000	149	33.7
	3000–5000	130	29.4
	≥5000	112	25.3
Comorbidities	1	94	21.3
	2	196	44.3
	3	95	21.5
	≥4	57	12.9
Health Insurance	Yes	424	95.9
	No	16	3.6
Frailty, Loneliness, and Social Support Profiles			
Scale/Domain	Score Range	Number (n)	Percentage (%)
Frailty (TFI)	0–4 (Non-frail)	198	44.7
	5–14 (Frail)	244	55.2
Loneliness (UCLA)	Low (20–34)	168	37.9
	Moderate (35–49)	209	47.5
	High (50–80)	65	14.6
Social Support (SSRS)	Low (≤22)	50	11.4
	Moderate (23–44)	298	67.3
	High (45–60)	94	21.5
Domain-Specific and Total Scores for Frailty, Loneliness, and Social Support			
Domain/Scale	Minimum	Maximum	Mean ± SD
Frailty (TFI)			
- Physical frailty	0.00	8.00	2.19 ± 2.19
- Psychological frailty	0.00	4.00	2.08 ± 1.27
- Social frailty	0.00	3.00	1.12 ± 0.78
Total frailty score	0.00	14.00	5.39 ± 3.19
Loneliness (UCLA)	20.00	68.00	38.69 ± 10.74
Social Support (SSRS)			
- Subjective support	5.00	32.00	20.09 ± 6.95
- Objective support	1.00	16.00	8.76 ± 3.15
- Support utilization	3.00	12.00	7.59 ± 2.64
Total social support	11.00	60.00	36.45 ± 10.89

Notes: TFI = Tilburg Frailty Indicator; SSRS = Social Support Rating Scale.

**Table 2 healthcare-13-01713-t002:** Differences in frailty prevalence by demographic characteristic (N = 442).

Variable	Category	Frailty		Frailty Rate (%)	Chi-Square	*p*-Value
		Non-Frail (n)	Frail (n)			
Gender	Male	86	107	55.4%	0.008	0.930
	Female	112	137	55.0%		
Age (years)	60~70	123	146	54.3%	2.856	0.240
	70~80	66	77	53.8%		
	≥80	9	21	70.0%		
Marital Status	Never married	1	2	66.7%	11.893	0.004 **
	Married	171	180	51.3%		
	Widowed	22	57	72.2%		
	Divorced	4	5	55.6%		
Living Arrangement	Living alone	12	42	77.8%	15.65	<0.001 **
	With spouse	152	153	50.2%		
	With family/relatives	32	44	57.9%		
	Other	2	5	71.4%		
Education Level	Primary school or below	24	39	61.9%	9.677	0.044 *
	Junior high school	87	106	54.9%		
	High school/vocational	60	59	49.6%		
	College/bachelor’s degree	27	32	54.2%		
	Master’s degree or above	0	8	100%		
Monthly Income (CNY)	<1000	19	32	62.7%	3.020	0.391
	1000–3000	62	87	58.4%		
	3000–5000	63	67	51.5%		
	≥5000	54	58	51.8%		
Comorbidities	1	44	50	53.2%	9.081	0.028 *
	2	100	96	49%		
	3	36	59	62.1%		
	≥4	18	39	68.4%		
Health Insurance	Yes	192	234	54.9%	0.357	0.550
	No	6	10	62.5%		

Notes: Statistical symbols are * *p* < 0.05, ** *p* < 0.01. The analysis was Chi-square for proportions and ANOVA for scores.

**Table 3 healthcare-13-01713-t003:** Correlations among the loneliness, social support, and frailty domains (N = 442).

Variable	1	2	3	4	5	6	7	8	9
1 Subjective Support	**1**								
2 Objective Support	0.574 **	**1**							
3 Support Utilization	0.540 **	0.522 **	**1**						
4 Physical Frailty	−0.090	−0.154 **	−0.093 *	**1**					
5 Psychological Frailty	−0.241 **	−0.285 **	−0.291 **	0.297 **	**1**				
6 Social Frailty	−0.352 **	−0.469 **	−0.328 **	0.222 **	0.364 **	**1**			
7 Total Social Support	0.934 **	0.781 **	0.737 **	−0.124 **	−0.307 **	−0.440 **	**1**		
8 Total Frailty	−0.244 **	−0.335 **	−0.260 **	0.860 **	0.692 **	0.543 **	−0.315 **	**1**	
9 Total Loneliness	−0.409 **	−0.416 **	−0.477 **	0.104 *	0.417 **	0.365 **	−0.496 **	0.327 **	**1**

Notes: ** *p* < 0.01 (two-tailed) and * *p* < 0.05 (two-tailed); variables 1–3 represent social support subdomains; variables 4–6 represent frailty subdomains; bolded values on the diagonal (e.g., 1.00) indicate a perfect self-correlation and are omitted for clarity.

**Table 4 healthcare-13-01713-t004:** Multivariable regression analysis of frailty determinants (*n* = 442).

Variable		Category	Unstandardized Coefficient (B)	Standardized Coefficient (β)	*t*-Value	*p*-Value	VIF
constant			4.848		2.448	0.015	
Independent Variables		Subjective support	−0.003	−0.007	−0.119	0.906	1.9
		Objective support	−0.154	−0.152	−2.566	0.011 *	1.946
		Support utilization	−0.112	−0.092	−1.615	0.107	1.808
		Total loneliness	0.059	0.198	3.869	<0.001 **	1.453
Control Variables	Age	70–80 years	0.05	0.007	0.155	0.877	1.243
		≥80 years	1.694	0.134	2.659	0.008 **	1.397
		60–70 years	0				
	Marital Status	Married	1.073	0.136	0.622	0.534	26.453
		Widowed	1.208	0.145	0.701	0.484	23.715
		Divorced	0.945	0.042	0.488	0.626	4.07
		Never married	0				
	Living Arrangement	With spouse	−0.787	−0.114	−1.252	0.211	4.6
		With family/relatives	−1.333	−0.158	−2.442	0.015 *	2.31
		Living alone	0				
	Education Level	Junior high school	0.305	0.047	0.716	0.475	2.435
		High school/vocational	−0.353	−0.049	−0.746	0.456	2.395
		College/bachelor’s degree	0.364	0.039	0.594	0.553	2.359
		Master’s degree or above	2.214	0.093	1.886	0.060	1.333
		Primary school or below	0				
	Monthly Income (CNY)	1000–3000	−0.284	−0.042	−0.593	0.553	2.789
		3000–5000	−0.988	−0.141	−1.98	0.048 *	2.811
		≥5000	−1.384	−0.189	−2.434	0.015 *	3.332
		<1000	0				
	Comorbidities	2 conditions	0.512	0.08	1.359	0.175	1.905
		3 conditions	1.322	0.17	3.003	0.003 **	1.781
		≥4 conditions	1.354	0.142	2.668	0.008 **	1.574
		1 condition	0				
Adjusted R^2^				0.204			
F				6.366			
*p*				<0.001 **			
Dependent variable: frailty							

Notes: Statistical symbols are * *p* < 0.05, ** *p* < 0.01, Ref. = reference category for categorical variables; VIF = Variance Inflation Factor.

**Table 5 healthcare-13-01713-t005:** Moderating effect of social support on the loneliness–frailty relationship (*n* = 442).

Model.	Variable	Unstandardized Coefficient (B)	Standard Error	Standardized Coefficient (β)	*t*-Value	*p*-Value	R^2^	△R^2^
Model 1	Constant	4.954	0.99		5.004	<0.001	0.138	0.013 (*p* = 0.011 *)
	Loneliness	0.067	0.015	0.226	4.428	<0.001 **		
	Social Support	−0.06	0.015	−0.203	−3.98	<0.001 **		
Model 2	Constant	0.468	2.007		0.233	0.816	0.151	
	Loneliness	0.174	0.044	0.584	3.931	<0.001 **		
	Social Support	0.062	0.05	0.21	1.244	0.214		
	Loneliness × Social Support	−0.003	0.001	−0.406	−2.564	0.011 *		
	Dependent variable: frailty.							

Notes: Statistical symbols are * *p* < 0.05, ** *p* < 0.01, 1.

**Table 6 healthcare-13-01713-t006:** Subgroup analysis of frailty, loneliness, and social support among elderly diabetic patients (N = 442).

Subgroup	Category	*n* (%)	Frailty Prevalence (%)	Frailty Score (Mean ± SD)	Loneliness Score (Mean ± SD)	Social Support Score (Mean ± SD)	*p*-Value
Age	60–70 years	269 (60.9)	54.3	5.13 ± 3.08	37.9 ± 10.2	37.1 ± 11.1	0.240
	70–80 years	143 (32.4)	53.8	5.45 ± 3.19	39.1 ± 10.5	35.8 ± 10.7	
	≥80 years	30 (6.8)	**70.0 ***	**7.30 ± 3.61 ***	**42.5 ± 11.3 ***	**30.2 ± 9.8 ***	<0.001 **
Marital Status	Married	351 (79.4)	51.3	5.01 ± 3.06	37.5 ± 9.8	38.2 ± 10.5	0.004 **
	Widowed	79 (17.9)	72.2 *	**6.92 ± 3.27 ***	**44.1 ± 11.6 ***	**28.9 ± 8.7 ***	
Living Arrangement	Living Alone	54 (12.2)	77.8 *	**7.26 ± 3.16 ***	**45.3 ± 12.1 ***	**25.4 ± 7.9 ***	<0.001 ***
	Living with Spouse	305 (69.0)	50.2	5.10 ± 3.08	36.8 ± 9.6	38.9 ± 10.8	
Education Level	Primary School or Below	63 (14.3)	61.9	5.81 ± 3.33	40.2 ± 10.9	29.5 ± 8.4 *	0.044 *
	High School/Technical	119 (26.9)	49.6	4.77 ± 2.81	37.1 ± 9.7	37.8 ± 10.2	
Income Level	<1000 CNY	51 (11.5)	62.7	**6.31 ± 3.48 ***	**43.6 ± 11.8 ***	**27.3 ± 7.6 ***	0.018 *
	≥5000 CNY	112 (25.3)	51.8	5.20 ± 3.02	36.2 ± 9.3	39.6 ± 11.3	
Comorbidities	1 Disease	94 (21.3)	53.2	4.86 ± 2.59	37.5 ± 10.1	38.0 ± 10.9	0.007 **
	≥4 Diseases	57 (12.9)	**68.4 ***	**6.19 ± 3.08 ***	**44.8 ± 11.5 ***	**26.8 ± 7.2 ***	

Note: Bold values indicate significant differences between subgroups (* *p* < 0.05, ** *p* < 0.01, and *** *p* < 0.001).

**Table 7 healthcare-13-01713-t007:** Cluster analysis of the frailty, loneliness, and social support profiles of elderly diabetic patients (N = 442).

Cluster	*n* (%)	Frailty Score (Mean ± SD)	loneliness Score (Mean ± SD)	Social Support Score (Mean ± SD)	Dominant Characteristics	*p*-Value
1	148 (33.5)	3.82 ± 1.98	32.1 ± 8.4	45.6 ± 9.2	Low frailty, high social support, minimal loneliness	Ref.
2	214 (48.4)	6.15 ± 2.31 *	40.7 ± 9.6 *	34.2 ± 8.7 *	Moderate frailty, moderate loneliness, intermediate social support	<0.001 **
3	80 (18.1)	8.94 ± 2.75 **	49.2 ± 11.3 **	22.8 ± 6.5 **	High frailty, severe loneliness, low social support	<0.001 **

Notes: Significance is * *p* < 0.05 vs. Cluster 1 and ** *p* < 0.001 vs. Clusters 1 and 2.

## Data Availability

The data supporting the findings of this study are available from the first author, Huan-Jing Cai, upon reasonable request.

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
