# Peer review of "Social Support’s Dual Mechanisms in the Loneliness–Frailty Link Among Older Adults with Diabetes in Beijing: A Cross-Sectional Study of Mediation and Moderation"

_healthcare, 2025, doi:10.3390/healthcare13141713_

Round 1

Reviewer 1 Report

Comments and Suggestions for Authors
  • Please improve the study title by adding the study location
  • Please emphasize the study novelty of the study in the abstract section
  • Please inform method section completely in the abstract section including the number of the participants.
  • Please improve this study problem statement in China. What makes China different compare to other country. Older adult in many country have the issue of loneliness and frailty. But, how about China? Why this issue should be conducted in China?
  • The participants number must be informed in the method section instead of introduction section.
  • The narration of sampling did not represent a multistage stratified sampling strategy at all.
  • The reference to calculate the sample size must be improved. The authors must include the statistical reference to calculate the participants number.
  • The reference of the prior validation of the instrument mist be provided.
  • Who collect the data? What is the criteria for the people who collect the data?
  • How to avoid recall bias, since the study using questionnaires instrument?
  • Tables must inform the number of thee participant, abbreviation information, statistical symbols information and statistical used for analysis.

Reviewer 2 Report

Comments and Suggestions for Authors

The manuscript reports findings of a cross-sectional analysis of the relationship between loneliness and frailty in a sample of Chinese older individuals with Type 2 diabetes, as well as the mediating and moderating effect of social support in the above association. This topic has high public health and clinical implications, yet there are some concerns that need some clarifications, as follows:

Title: The use of the phrase ‘dual pathways’ is a bit confusing and deserves a consistent definition throughout the manuscript. It somehow refers to the direct of loneliness on frailty and the indirect effect via social support (e.g., dual pathways: loneliness exacerbates frailty both indirectly through diminished social support and directly via independent mechanisms, lines 203-204); and also refers to mediating and moderating effects of social support (e.g., dual-pathway verification in diabetes-frailty research, lines 84 – 85). The use of that phrase needs to be standardized throughout the manuscript.

Introduction:

  • In line 65, the specific content and purpose of the ‘loneliness-health outcome model’ require further explanation.
  • In lines 68-70, the formulation of the ‘social support’s dual mediating and moderating role’ is insufficient. The theoretical and empirical foundations for both the mediating and moderating roles of social support should be extensively articulated. More importantly, mediation and moderation are statistical concepts; it is crucial to clearly discuss their theoretical and empirical significance within a single study. Since these approaches rest on distinct assumptions about the processes under investigation, this distinction should be explicitly explained to readers to strengthen the Introduction. This is  a major concern for this manuscript.
  • In lines 82-83, the relevance of the psychosocial stress-inflammation-functional decline model to this manuscript is not clear. Since inflammation is neither a focus of this study nor assessed, the argument linking to inflammation should be minimized here (and also throughout the manuscript).

Methods:

  • Please clarify the ‘major stressors’ (line 104) considered in this study.
  • On what rationales were levels of loneliness and social support categorized into low, moderate and high? Were these just for the purpose of the simple slope analysis, or descriptive statistics, as in Table 1)? And based on what criteria did the stated cut-offs use? Please justify.
  • The application of subgroup differences in demographic characteristics (Results 3.6) and identification of high-risk subgroups (Results 3.7) should be mentioned here. Please provide the rationales and details of these analyses here.

Results:

  • In lines 207- 208, the significant mediating effect is suggested to support ‘the critical role of social support as a modifiable buffer against loneliness-driven frailty progression in older adults with diabetes’. I suppose this is more related to the moderating, rather than the mediating effect, as the mediating effect aligns more with social support, as the underlying pathological mechanisms from loneliness to frailty.
  • Under subsection 3.5 where the results of the moderating effect of social support is discussed, please state how the low and moderate and high levels of social support were defined. Were the same cut-offs as mentioned under the Methods (ref line 119) used? Also, what values were the Ws represented here?
  • In Figure 2, it would be more complete to show the regression line for the moderate level of social support, to be aligned with the above results.

Discussion:

  • The discussion is overly descriptive, and its theoretical depth should be enhanced. More than half of the discussion focuses on the potential clinical impacts and intervention implications of the results, some of which are over-interpretations of cross-sectional findings and should be condensed. Importantly, the directionality of the associations in the mediation model should be interpreted more cautiously, as the possibility of reverse causality—from frailty to loneliness (which have been reported in some longitudinal studies—cannot be ruled out. Consequently, the intervention recommendations should be more careful, selective and concise.
  • Various theories/ hypotheses appeared here to explain the results, which could have been introduced earlier in the Introduction to set the stage of the entire investigation.
  • In addition, it is a bold statement to state that ‘the study’ innovation lies in quantifying intervention thresholds’ (line 456), as there is a huge knowledge gap from the current cross-sectional results to establishing its potential for intervention. The following statements in this paragraph contain quite lot of jargon and the meanings are not clear for comprehension. Please rewrite for clarity.
  • It is not straightforward to see how wearables could contribute to the current understanding of the relationship between loneliness, frailty and social support. What parameters from wearables (and which wearables) would facilitate our understanding of this relationship? Please elaborate.

Reviewer 3 Report

Comments and Suggestions for Authors

Dea Authors,

the comments in the annex file.

Best

Author Response

The STROBE-checklist is the supplementary materials

Round 2

Reviewer 2 Report

Comments and Suggestions for Authors

As noted in my initial review, the manuscript examined cross-sectional relationship between loneliness and frailty in a sample of Chinese older individuals with Type 2 diabetes, as well as the mediating and moderating effect of social support in the above association. The authors are commended for providing a thorough response to my and another reviewer’s feedback, particularly in clarifying the formulation of the research question, methodology and results. I would like the authors to further clarify the revised statement "Mediation implies support is a causal intermediary; moderation implies it alters loneliness-frailty strength. Our design tests both hypotheses separately using Hayes’ PROCESS framework." In particular, it would be crucial to elaborate further on how social support could be a mediator and moderator considered in a single study, as both take on distinct assumptions of the nature of social support. If social support is assumed to fluctuate with loneliness and frailty, then social support should behave like a mediator. On the other hand, if social support is assumed to be stable across time, it should behavior more like a moderator. Indeed, in reality, it could be complicated for a clear-cut differentiation, and this could be elaborated more to set the stage for BOTH analyses in this study. I would also suggest removing the part on ‘PROCESS framework’, as statistically speaking, both approaches are feasible, and this shouldn’t be confused with the conceptual understanding of social support here. The rest of the manuscript looks good to me.

Reviewer 3 Report

Comments and Suggestions for Authors

Dear Auhors,

in this form the manuscript certly more valid but topic suggested for support the part of clinical practice view not are completed and update with reference more recent and valid. Support with reference more recent suggested for the topic “Lifestyle Medicine Case Manager Nurses for Type Two Diabetes” and “Nurse-Led Care Coordination in a Transitional Care in Patients With Diabetes”. Please update the native review the manuscript, principally in new part addded. Best

Comments on the Quality of English Language

Merit major attention, principally in part updated
